# Subjective Satiety Following Meals Incorporating Rice, Pasta and Potato

**DOI:** 10.3390/nu10111739

**Published:** 2018-11-12

**Authors:** Zhuoshi Zhang, Bernard J. Venn, John Monro, Suman Mishra

**Affiliations:** 1Department of Human Nutrition, University of Otago, P.O. Box 56, Dunedin 9054, New Zealand; zhuoshi.zhang@gmail.com (Z.Z.); bernard.venn@otago.ac.nz (B.J.V.); 2New Zealand Institute for Plant & Food Research, Private Bag 11600, Palmerston North 4442, New Zealand; John.Monro@plantandfood.co.nz

**Keywords:** carbohydrate, satiety, mixed meal, potato, pasta, rice

## Abstract

The satiating capacity of carbohydrate staples eaten alone is dependent upon the energy density of the food but relative satiety when starchy staples are incorporated into mixed meals is uncertain. Our aim was to assess the satiating effects of three carbohydrate staples; jasmine rice, penne pasta, and Agria potato, each consumed within a standard mixed meal. Cooked portions of each staple containing 45 g carbohydrate were combined with 200 g of meat sauce and 200 g of mixed vegetables in three mixed meals. The quantities of staple providing 45 g carbohydrate were: Rice, 142 g; pasta, 138 g and potato 337 g. Participants (*n* = 14) consumed each of the mixed meals in random order on separate days. Satiety was assessed with using visual analogue scales at baseline and for 3 h post meal. In an area-under-the-curve comparison, participants felt less hungry (mean (SD)) following potato 263 (230) than following rice 374 (237) or pasta 444 (254) mm∙min, and felt fuller, more satisfied, and wanted to eat less following the potato compared with the rice and pasta meals (*p* for all <0.01). The superior satiating effect of potato compared with rice and pasta in a mixed meal was consistent with its lower energy density.

## 1. Introduction

The World Health Organization estimates that more than half of the world’s adults are overweight or obese and warns of dramatic rises in overweight and obesity in low- and middle-income countries [1]. Overweight and obesity statuses are multifactorial in aetiology with such factors as the food environment, decreased physical activity, inadequate sleep and medication being involved [2,3]. Highly palatable foods that are hard to resist eating are said to be ‘hyperpalatable’ and are characterized as being inexpensive, highly caloric, fat-laden, potentially addictive and a major contributor to chronic weight gain [4,5]. The involvement of high calorie foods as a contributing factor to overweight and obesity has some credibility as food energy density has been associated with body weight change [6]. A strategy used to reduce dietary energy density has been to add pureed (hidden) vegetables into meals, resulting in a reduction in energy intake [7]. However, the effectiveness of adding fruit and vegetables into diets as a means of reducing overall dietary energy density is questionable as a meta-analysis of the body of literature indicated no long-term effect on weight loss [8]. This may be due to foods such as non-starchy vegetables having a low impact on feelings of satiety when eaten in typical amounts [9]. The authors of the meta-analysis concluded that increasing the fruit and vegetable intake of a diet is unlikely to be a successful weight control strategy because people tend not to decrease their overall dietary energy intake, suggesting that mechanisms underlying participant perceptions of hunger and satiety need to be better understood [8].

At a population level, low energy dense diets of good nutritional quality in Irish children and teenagers have been characterized as having higher intakes of fruits, vegetables, grains, rice, pasta and potato (boiled, baked, mashed); and lower intakes of carbonated beverages and chipped, fried and roasted potatoes [10]. The consumption of low energy dense foods must be accompanied with a concomitant reduction in the intake of high energy dense foods in order for an effect on body weight to be observed [8]. Indeed, the energy density of diets, characterized by a combination of lower fat and higher fruit and vegetable intake, have been associated with long-term weight loss [11]. In an ad libitum dietary intervention in which fat was replaced with either starch- or sugar-rich foods, spontaneous weight loss was achieved with the starchy diet over 2 weeks with the authors suggesting that the starch-based diet was more satiating than the higher fat or sugar diets [12]. Satiety is an important aspect to consider when changing the energy density of meals. When foods were consumed in servings containing 1000 kJ, satiety was inversely related to the energy density of the food, with a strong positive correlation found between the satiety index of food and the serving weight (*r* = 0.66, *p* < 0.001) [13]. The least satiating foods were energy-dense snacks, confectionary and high-fat bakery products; among the starchy foods, white rice and pasta had satiety indices 1.19 and 1.38 times higher than white bread, respectively, with potato having a satiety index over three times (3.23) that of white bread [13].

This evidence is indicative that potato may be of research interest in relation to satiety. In a comparison of instant mashed potato and barley containing 49.5 g and 46.6 g carbohydrate, respectively, there was no difference in satiety between foods although half of the 10 test participants could not finish all of the food [14]. However, people seldom eat single foods so the incorporation of starchy foods into test meals may have more practical relevance. In one such study, 12 participants consumed breakfast meals that included 50–52 g available carbohydrate in the form of baked potato, instant potato, brown rice, white bread and pasta [15]. There was no difference in area-under-the-curve (AUC) hunger or fullness ratings among the meals, with a possible explanation being that a cup of water of variable volume accompanied the food such that the overall water content of the meals plus drink was standardized to 400 mL [15]. This would have evened out differences in energy density among the meals and may have obscured the previously observed effect of energy density on satiety [13]. Therefore, although previous studies are suggestive of a differential effect of starchy foods on satiety, the data are inconsistent, possibly as a consequence of study design. We hypothesized that preserving differences in energy density among meals would maintain differences in feelings of satiety.

Thus, we provided cooked meals for lunch, a usual time for such food to be served, accompanied with a fixed volume of water to drink in order to preserve differences in energy density among meals. Our aims were to assess, in a realistic and normal lunchtime setting, the immediate effect on satiety following meals containing the starchy staple foods; rice, pasta and potato, as part of a mixed meal.

## 2. Materials and Methods

### 2.1. Experimental Design

A randomized crossover design was used. On three different days, participants consumed a lunchtime meal comprising minced beef in a Bolognese sauce, mixed vegetables, and a serving of either whole boiled potato, white rice or penne pasta. The order in which participants received the meals was randomized and there was a minimum two-day washout between meals. Participants were asked to keep breakfast consistent on each of the three test days, to avoid consuming any food between breakfast and lunch, and to avoid any food intake for three hours after finishing the test meal, the period during which satiety was measured.

### 2.2. Recruitment

Volunteers were recruited using a flyer that briefly described the study. In advance of the study each respondent was presented with an information sheet and an informed consent form. Fourteen healthy volunteers (nine females and five males) were recruited using the following inclusion/exclusion criteria.

### 2.3. Inclusion Criteria

Age: Aged between 18 and 70.

Sex: Male or female.

Gastrointestinal function: No history of gastrointestinal dysfunction that could have an impact on appetite in the three hours after consuming the meal.

Health: Healthy as gauged by self-assessment and results on the General Health Questionnaire.

Activity: Not involved in prolonged strenuous activity on the day of the trial

Agreement: Subject having given written informed consent to comply with the conditions of the trial.

### 2.4. Exclusion Criteria

Self-reported intolerance to any of the meal components.

Having a gastrointestinal disorder.

Being involved in a physically demanding work on the day of the trial.

Eligible volunteers who were willing to participate were invited to attend the study at the Plant & Food premises.

Ethical approval was obtained from the New Zealand Health and Disabilities Ethics Committee (HDEC 15/CEN/71). This study was registered on the Australian New Zealand Clinical Trials Registry (www.anzctr.org), registration number ACTRN12615000721505. All participants signed a consent form.

### 2.5. Test Meal Preparation and Composition

All meals were prepared in advance, frozen as individual meals and reheated before consumption. The carbohydrate foods were cooked and served in equal carbohydrate and calorie portions, and consumed with the standardized meat and vegetable sauce, and 250 mL of water. The meat sauce contained fried beef mince, bacon, onions, chopped celery, thyme, bay leaves and chopped black olives. A commercial pasta sauce (Dairymaid foods, Christchurch, New Zealand) was added and the whole mixture stirred and simmered for 90 min. The mixed vegetables were a commercial frozen mixture consisting of peas, chopped carrots and corn (Heinz Wattie’s Ltd., Hastings, New Zealand) and were boiled for 5 min. The pasta (penne; Diamond brand, Wilson Consumer Products, Auckland, New Zealand), rice (Jasmine; SunRice, New South Wales, Australia) and whole potatoes (Agria; Morgan Laurenson Ltd., Palmerston North, New Zealand) were weighed, cooked according to package instructions, and reweighed after cooking. The meals were assembled by placing 200 g vegetables, 200 g Bolognese sauce, and the weight of the rice, penne pasta or whole potato containing 45 g of carbohydrate (Table 1) into aluminum containers which were sealed and frozen until required. Before consumption, the meals were thawed overnight in a refrigerator and heated in a convection oven.

On test days each participant was asked to consume his or her normal breakfast and to refrain from eating between 8 a.m. and 12.00 p.m. (lunch time). Participants were asked to consume a similar breakfast on each of the 3 test days. At lunch time the participants were provided with the test meal and asked to consume it within 20 min.

### 2.6. Satiety

Satiety was measured using a 100 mm visual analogue scale (VAS) consisting of four questions anchored at either end with the following statements:How hungry do you feel? (Not at all hungry–Extremely hungry)How full do you feel? (Not at all full–Totally full)How strong is your desire to eat? (Not at all strong–Extremely strong)How much food do you think you can eat? (Nothing at all–A large amount)

These scales have been recommended in a methodological review of the evaluation of foods for their validity and reliability [16]. The length of the scale was measured from the start to the point that was marked. Participants were asked to rate their hunger/appetite using the VAS immediately before lunch, immediately after lunch and at 1, 2 and 3 h after lunch. The VAS rating for each time was on a separate sheet and participants were instructed not to refer back to ratings of earlier times. 

### 2.7. Sample Size Calculation

In a study on the validity of appetite visual analogue scales, a Table was presented in which sample sizes were given in relation to detectable differences and power [17]. Using this information, 18 subjects would be sufficient to detect a 10% difference in satiety, with 80% power at the 0.05 significance level for a paired design. 

### 2.8. Statistical Analysis

All appetite ratings were recorded in a spreadsheet using the Microsoft Excel for Macintosh (Microsoft^®^ Excel^®^. Version 15.31. Microsoft Corporation 2017, Redmond, WA, USA). Area under the curve (AUC) was calculated. Results were expressed as means with standard deviation. Random effects regression analysis was used to test for between-treatment differences in AUC satiety responses, with participant id as a random effect and adjusted for randomized order and baseline satiety. Microsoft Stata/MP 14.0 for Macintosh (Stata Corporation, College Station, TX, USA) was used for the regression analysis. *p*-value of less than 0.05 was set as statistical significance in all analyses.

## 3. Results

Fourteen adults completed the intervention in a balanced three-arm crossover. The mean (SD) age of the participants was 40.9 (14.6) years with a range of 28–70 years. Eleven participants were of European- and 3 of Asian-descent. The flow of participants through the study is given in Figure 1.

The baseline scores to the satiety questions are given in Table 2. People were randomized to the order in which they received the meals. 

At baseline, there was no difference in mean scores of hunger and fullness before eating. There was a lesser desire to eat and people indicated they could eat less on the potato and rice days compared with the pasta day. The main outcomes arising from the visual analogue scale data are given in Table 3. The outcomes are total postprandial AUC over three hours; the difference (mm) between the pre- and post-meal scores (0–15 min); and a rate of return to hunger (15–180 min).

The plots of visual analogue scale data over time are shown in Figure 2.

## 4. Discussion

The study results are indicative that the rice and the pasta meals were equally satiating, whilst participants felt fuller and more satisfied after eating the potato meals compared with the rice and the pasta meals. In this experiment, the comparison among different carbohydrate foods was standardized to an equal carbohydrate content. Similar results have been found when testing foods on an isoenergetic basis in which boiled potatoes eaten alone were more satiating than either rice or pasta eaten alone [13]. The data from the present study in a meal setting, are therefore consistent with differences in satiety found when these foods are eaten alone.

Our findings may be compared with a study in which investigators tested the effects on satiety of consuming meals containing various starchy staples [15]. In that study, desire to eat was lower following baked potato compared with pasta but contrary to our findings, hunger and fullness AUC did not differ among meals containing instant potato, baked potato, brown rice, pasta and white bread [15]. A difference between study designs was that Geleibter et al. standardized the total water content of the meals (food plus water provided as a beverage) whereas we only standardized the volume of water given as an accompanying beverage. Thus, the meals and beverages that Geleibter et al provided presumably tended to be equi-volumetric [15]. This may have obscured a difference in satiety among the meals given that food volume is a determinant of satiety [18]. This is suggestive that if starchy foods are to be exchanged, keeping accompanying drink volume constant may be important when assessing effects on satiety.

With comparable protein, fat, dietary fiber and calorie content, possible explanations for potato being more satiating than rice or pasta are differences in the energy density of the foods. Potatoes have a higher water content and lower energy density than rice or pasta [19]. Therefore, a larger volume of potatoes than rice and pasta needed to be consumed when served in equal carbohydrate portions. A large food volume increases gastric distension and stimulates postprandial satiety [20,21,22]. When eaten ad libitum, potato, rice and pasta consumed with a pork steak resulted in satiating effects that were not different among the meals despite the total carbohydrate and calorie intake after eating potatoes being significantly less than after eating rice and pasta meals [23]. In a randomized crossover study involving 11 to 13-year-old children, 30–40% less calories (*p* < 0.0001) were consumed after eating ad libitum meals containing boiled and mashed potatoes compared with comparable pasta and rice meals [24]. These observations may have practical and clinical relevance because the amount of carbohydrate consumed directly impacts postprandial glycaemia. If satiety can be maintained whilst consuming a smaller portion of potato compared to other starchy staples, this may offset the potential for a larger glycemic excursion due to potato having a high glycemic index (GI) relative to rice and pasta [25]. A consequence of a smaller portion reduces not only the glycemic load (GI x grams available carbohydrate) but also the energy content of the meal. As suggested, potatoes could be a suitable food option to reduce energy intake whilst maintaining satiety and mitigating postprandial glycaemia as less carbohydrate is consumed [26]. This effect has been observed among an older group consuming an equivalent amount of carbohydrate either as a glucose beverage, instant potato or barley [14]. All 10 participants were able to ingest the beverage but four and five of the participants were unable to eat all of the potato and barley, respectively [14]. Despite the differences in carbohydrate intake of the subset who could not finish their food, as a group mean satiety was greater after potato than after the glucose beverage.

It is interesting to note that the satiating effect of the carbohydrate foods was not clearly related to GI, even though it has been proposed that GI, as an indicator of sustained carbohydrate digestion, is a strong determinant of satiety [27,28]. Potato and jasmine rice are generally considered to be of moderate to high GI, and pasta of low GI, yet the rice and pasta meals did not differ in satiation and the potato meal was more satiating than the pasta meal. The results suggest that the effect of food volume on the proportion of a meal released from the stomach per unit time may have been sufficient to override the effects of differences in digestibility of the carbohydrates. Indeed, the rate of gastric emptying has been found to be affected by food volume and by energy density [29]. There is also a relationship between glycaemia and gastrointestinal motor control indicative that hyperglycemia slows gastric emptying [30]. More research in which blood glucose responses and gastric emptying are measured in conjunction with satiety would be helpful in interpreting the results of the present study.

It has also been found that although the glycemic response to mashed potato was greater compared with rice and pasta when consumed as part of a meal containing vegetables and salmon, the glycemic responses to all three meals were not different [31]. If the same effect has occurred in the present study due to the presence of meat sauce and vegetables, the effects of differences in glycemic response on satiety may have been eliminated leaving food volume as a dominant influence on satiety.

The healthfulness of carbohydrates in the human diet has been examined from a migratory perspective in which carbohydrate-rich staple foods consumed in the country of birth have been replaced by an increased intake of refined carbohydrate, meat and dairy in the adopted country contributing to a higher risk of non-communicable disease [32]. The rice and pasta used in our study were refined and perhaps equivalent whole-grain products would have induced different satiety responses. In previous work there was no difference in satiety found between white and brown rice; brown pasta appeared to be more satiating than white pasta; and boiled potatoes were the most satiating of all of the foods tested [13]. Plant-based starchy and non-starchy foods in general have lower energy density than animal derived foods and foods that are highly possessed with sugar and fat [19]. In practice, a plant-based diet in which starchy foods were recommended to be eaten ad libitum to satiation resulted in a greater reduction in body mass index compared with a usual care control group [33]. Thus, there is evidence to suggest that low energy-dense starchy foods are healthful components of a diet even when eaten ad libitum to satiation. 

A potential limitation to our work was the sample size. The study had been powered to detect a difference in satiety among meals and it was sufficient for this purpose. Also, there were no differences at baseline in VAS responses to the questions “How hungry do you feel?” and “How full do you feel?”. However, the data were indicative that people had a greater desire to eat; and could eat a larger quantity before the pasta meal compared with the rice and potato meals. We are unsure why this difference occurred as the order in which participants received the meals was randomized and we would not have expected baseline differences to occur. It is unclear whether this is a sample size issue, a chance finding, or whether anticipatory effects could have caused such an outcome. VAS methodology is widely used for assessing satiety [16] but subsequent energy consumption and satiety hormone responses could be informative as objective measurements in future research. A strength of this study was that all test meals were realistic in composition. It provides evidence that reflects real life when meat and vegetables are consumed in combination with starchy staples. A limitation to generalization of this study was that it was not conducted in people who are overweight or obese, a demographic who may have an impaired satiety response [34]. Similar studies undertaken in groups of people who are overweight or obese could provide evidence on which to base dietary recommendations suitable for weight loss.

## 5. Conclusions

In summary, on an equal carbohydrate basis, potato meals were more satiating than rice or pasta meals. If serving sizes of potato could be reduced such that the satiating properties match that of larger servings of rice or pasta, there would be a caloric intake saving and the potential to bring the glycemic response of high GI potato down to the glycemic responses of the rice and pasta meals, making potatoes an excellent choice of a low energy-dense food with the capacity to satiate. 

## Figures and Tables

**Figure 1 nutrients-10-01739-f001:**
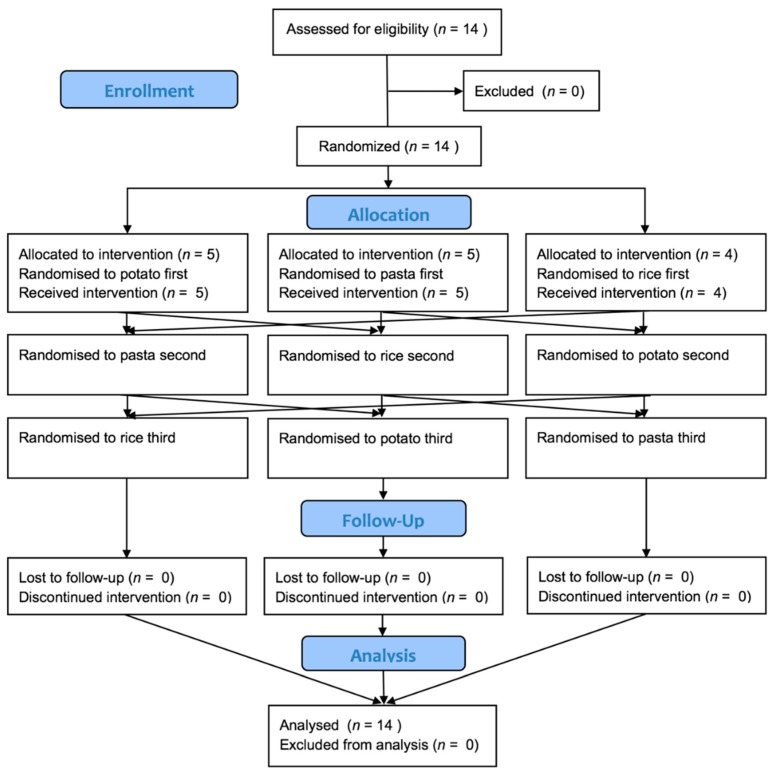
CONSORT diagram showing the flow of participants through the study.

**Figure 2 nutrients-10-01739-f002:**
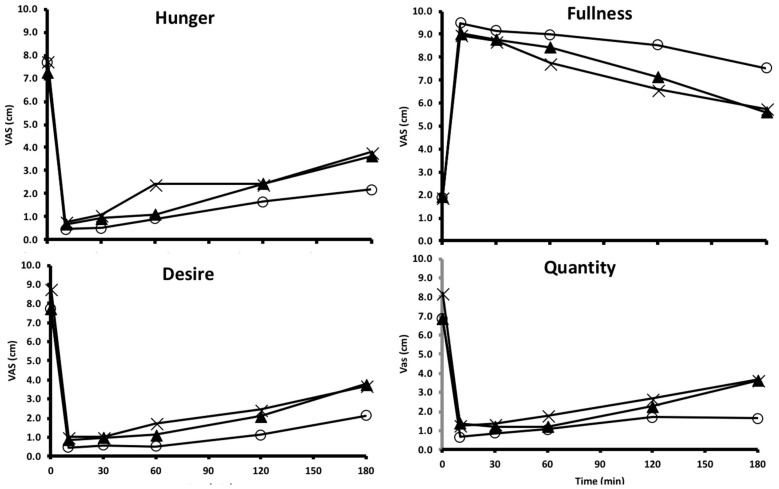
Plots of visual analogue scales (VAS) in response to four questions over time starting at baseline (t = 0) and following the consumption of pasta (×), potato (○) and rice (▲) meals. Data were analyzed by comparing the area-under-the-curve (AUC) among the meals for each of the questions. Interpretation of the data: How hungry do you feel? (Small AUC—Not at all hungry; Large AUC—Extremely hungry). How full do you feel? (Small AUC—Not at all full; Large AUC—Totally full). How strong is your desire to eat? (Small AUC—Not at all strong; Large AUC—Extremely strong). How much do you think you can eat? (Small AUC—Nothing at all; Large AUC—A large amount).

**Table 1 nutrients-10-01739-t001:** Test meal composition.

Meal	Meal Components			Total CHO (g)	Total Energy (kJ)	Energy Density (kJ/g)
	Mince + Sauce (g)	Vegetables (g)	Starchy Staple (g)			
Rice	200	200	142	112	3010	5.55
Pasta	200	200	138	112	3010	5.59
Whole potato	200	200	337	112	3010	4.08

**Table 2 nutrients-10-01739-t002:** Mean (SD) baseline satiety scores of 14 people.

Satiety Measure (mm) *	Pasta	Potato	Rice
Hunger	77.4 (21.9)	77.1 (10.2)	72.9 (19.8)
Fullness	18.9 (16.9)	18.7 (12.1)	19.6 (14.9)
Desire	87.9 (11.1) ^a^	77.1 (12.5) ^b^	77.3 (19.1) ^b^
Quantity	81.9 (11.4) ^a^	68.9 (11.2) ^b^	68.6 (19.8) ^b^

Different superscript letters within a row signify statistically significant. differences. * A high score indicates hunger; fullness, desire to eat; and ability to eat a large quantity.

**Table 3 nutrients-10-01739-t003:** Mean visual analogue scale (VAS) outcomes of 14 people in response to four satiety questions.

Satiety Measure ^1^	Mean (SD)			Mean Difference (95% Confidence Interval) Comparing between Meals Given in the Column Headings above		
	Pasta	Potato	Rice	Pasta vs. Potato	Pasta vs. Rice	Rice vs. Potato
Hunger AUC	448 (254)	264 (230)	376 (236)	184 (105, 263) ***p* < 0.001**	−72 (−151, 8) *p* = 0.076	112 (33, 191) ***p* = 0.006**
Fullness AUC	964 (468)	1120 (345)	954 (423)	−155 (−335, 24) *p* = 0.090	−10 (−189, 170) *p* = 0.914	−165 (−344, 14.3) *p* = 0.071
Desire AUC	427 (220)	210 (190)	361 (218)	217 (147, 287) ***p* < 0.001**	−66 (−136, 4) *p* = 0.064	151 (87, 214) ***p* < 0.001**
Quantity AUC	436 (231)	289 (257)	389 (221)	148 (65, 230) ***p* < 0.001**	−47 (−130, 36) *p* = 0.265	100 (30, 171) ***p* = 0.005**
Change (mm) in VAS scores from commencement of eating to finishing the meal						
Hunger drop ^2^	76 (18)	72 (15)	71 (16)	3.6 (−2.5, 9.8) *p* = 0.249	−4.6 (−10.9, 1.6) *p* = 0.146	−1.0 (−7.2, 5.2) *p* = 0.755
Fullness rise ^2^	70.6 (18.3)	75.7 (13.3)	71.4 (19.7)	−5.1 (−8.0, −2.2) ***p* = 0.001**	0.7 (−2.1, 3.7) *p* = 0.617	−4.3 (−7.2, −1.4) ***p* = 0.004**
Desire drop	87.9 (11.1)	77.1 (12.5)	77.3 (19.1)	10.9 (2.3, 19.4) ***p* = 0.013**	−10.6 (−19.2, −2.1) ***p* = 0.015**	0.2 (−8.4, 8.8) *p* = 0.961
Quantity drop	81.9 (11.4)	68.9 (11.2)	68.6 (19.8)	13.0 (5.2, 20.8) ***p* = 0.001**	−13.4 (−21.2, −5.5) ***p* = 0.001**	−0.4 (−8.1, 7.5) *p* = 0.929
Rate of return of VAS scores (mm/h) from eating cessation to 3 h post-baseline						
Hunger return ^3^	9.8 (6.5)	6.5 (9.0)	10.6 (9.3)	3.2 (−0.1, 6.6) *p* = 0.061	0.9 (−2.5, 4.3) *p* = 0.609	4.1 (0.7, 7.5) ***p* = 0.017**
Fullness return	−10.6 (7.0)	−6.3 (8.5)	−12.2 (9.4)	−4.2 (−8.3, −0.2) ***p* = 0.040**	−1.6 (−5.7, 2.4) *p* = 0.425	−5.9 (−9.9, −1.8) ***p* = 0.004**
Desire return	9.7 (6.1)	5.7 (9.6)	10.2 (8.8)	4.0 (−1.1, 9.1) *p* = 0.121	0.5 (−4.6, 5.6) *p* = 0.850	4.5 (−0.2, 9.2) *p* = 0.060
Quantity return	8.2 (3.9)	3.5 (3.6)	8.6 (6.2)	4.7 (1.5, 7.9) ***p* = 0.004**	0.4 (-2.8, 3.6) *p* = 0.803	5.1 (2.3, 7.9) ***p* < 0.001**

^1^ AUC Area-Under-the-Curve (cm∙min); ^2^ drop/rise = change in score (mm) from baseline to immediately after finishing the meal; ^3^ return from a satiated to a less satiated condition over time (mm/h). A positive difference in hunger, desire and quantity represent being hungrier, having greater desire and a feeling of being able to eat more. A negative difference in fullness represents feeling less full. Bolded *p*-values indicate differences between treatments.

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
