# Peer review of "Subjective Satiety Following Meals Incorporating Rice, Pasta and Potato"

_nutrients, 2018, doi:10.3390/nu10111739_

Reviewer 1 Report

1. Introduction: This section is too short in its current format. (1) The introduction could benefit from a more recent scan of the literature. The authors seem to have omitted literature relevance to this study. e.g. Geliebter et al (2013). Satiety following intake of potatoes and other carbohydrate test meals. (2) What is the theoretical background of the study? Many theories of appetite control have focused on the status of macronutrients (e.g. the glucostatic theory) (3) It would help to clarify the purpose of the study for the authors to state explicit hypotheses at the close of the introduction.

2.  Methods: A reference should be included for the validity and reliability of visual analogue scale (VAS).

3.  Methods: Please provide additional information on how you calculated your sample size.

Author Response

We thank the reviewer for insightful comments. We hope we have satisfactorily addressed these.

 Introduction: This section is too short in its current format.

Authors response: We have lengthened the Introduction based on the reviewers comments as follows. “This evidence is indicative that potato may be of interest to study in relation to satiety. In a comparison of instant mashed potato and barley containing 49.5g and 46.6g carbohydrate, respectively, there was no difference in satiety between foods although half of the 10 test participants could not finish all of the food {Kaplan, 2002 #23}. However, people seldom eat single foods so the incorporation of starchy foods into test meals may have more practical relevance. In one such study, 12 participants consumed breakfast meals that included 50 – 52g available carbohydrate in the form of baked potato, instant potato, brown rice, white bread and pasta {Geliebter, 2013 #24}. There was no difference in area-under-the-curve (AUC) hunger or fullness ratings among the meals, with a possible explanation being that a cup of water of variable volume accompanied the food such that the overall water content of the meals plus drink was standardized to 400 mL {Geliebter, 2013 #24}. This would have evened out differences in energy density among the meals and may have obscured the previously observed effect of energy density on satiety {Holt, 1995 #11}. Therefore, although previous studies are suggestive of a differential effect of starchy foods on satiety, the data are inconsistent, possibly as a consequence of study design. We hypothesized that preserving differences in energy density among meals would maintain differences in feelings of satiety. Thus, we provided cooked meals for lunch, a usual time for such food to be served, accompanied with a fixed volume of water to drink in order to preserve differences in energy density among meals. Our aims were to assess, in a realistic and normal lunchtime setting, the immediate effect on satiety following meals containing the starchy staple foods; rice, pasta and potato, as part of a mixed meal.

 The introduction could benefit from a more recent scan of the literature. The authors seem to have omitted literature relevance to this study. e.g. Geliebter et al (2013). Satiety following intake of potatoes and other carbohydrate test meals.

Authors response: Thank you, this is a very relevant reference, we have included this and another by Kaplan et al.

 What is the theoretical background of the study? Many theories of appetite control have focused on the status of macronutrients (e.g. the glucostatic theory). It would help to clarify the purpose of the study for the authors to state explicit hypotheses at the close of the introduction.

Authors response: We have included a hypothetical basis for the experiment in the revised Introduction.

 Methods: A reference should be included for the validity and reliability of visual analogue scale (VAS).

Authors response: We have included the following: “These scales have been recommended in a methodological review of the evaluation of foods for their validity and reliability {Blundell, 2010 #20}”.

  Methods: Please provide additional information on how you calculated your sample size.

Authors response:  We have added the following into the Methods: “In a study on the validity of appetite visual analogue scales, a Table was presented in which sample sizes were given in relation to detectable differences and power {Flint, 2000 #12}. Using this information, 18 subjects would be sufficient to detect a 10% difference in satiety, with 80% power at the 0.05 significance level for a paired design.”  

Reviewer 2 Report

The subject is very original and interesting, but it is presented in a very concisely manner. Generally The Introduction should be enlarged, the methods should be presented with major details and the results should be discussed with major argumentations. Also the authors should mark that this represents a pilot study. The linguistic revision of whole manuscript is needed.

In details:

lines 29-30 should be rewritten;

lines 30-32 should be clarified;

lines 34-37 should be rewritten;

lines 30-44 these lines should be  better linked among them;

lines 45-48 should be enlarged.

The aim of work should be clarified.

Subparagraph 2.1 should be clarified and rewritten.

The authors should explain the choice of typology of test meal.

The data in table 3 and Figure 2 should be better described.

In the discussion the data should better compared  with previous data in literature

Author Response

We thank the reviewer for insightful comments. We hope we have satisfactorily addressed these.

 The subject is very original and interesting, but it is presented in a very concisely manner. Generally

 The Introduction should be enlarged.

Authors response: Thank you, we have added the following: “This evidence is indicative that potato may be of interest to study in relation to satiety. In a comparison of instant mashed potato and barley containing 49.5g and 46.6g carbohydrate, respectively, there was no difference in satiety between foods although half of the 10 test participants could not finish all of the food {Kaplan, 2002 #23}. However, people seldom eat single foods so the incorporation of starchy foods into test meals may have more practical relevance. In one such study, 12 participants consumed breakfast meals that included 50 – 52g available carbohydrate in the form of baked potato, instant potato, brown rice, white bread and pasta {Geliebter, 2013 #24}. There was no difference in area-under-the-curve (AUC) hunger or fullness ratings among the meals, with a possible explanation being that a cup of water of variable volume accompanied the food such that the overall water content of the meals plus drink was standardized to 400 mL {Geliebter, 2013 #24}. This would have evened out differences in energy density among the meals and may have obscured the previously observed effect of energy density on satiety {Holt, 1995 #11}. Therefore, although previous studies are suggestive of a differential effect of starchy foods on satiety, the data are inconsistent, possibly as a consequence of study design. We hypothesized that preserving differences in energy density among meals would maintain differences in feelings of satiety. Thus, we provided cooked meals for lunch, a usual time for such food to be served, accompanied with a fixed volume of water to drink in order to preserve differences in energy density among meals. Our aims were to assess, in a realistic and normal lunchtime setting, the immediate effect on satiety following meals containing the starchy staple foods; rice, pasta and potato, as part of a mixed meal.

 The methods should be presented with major details.

Authors response: Based on your comments and those of another reviewer, we have added the following information into the Methods. “These scales have been recommended in a methodological review of the evaluation of foods for their validity and reliability {Blundell, 2010 #20}” and “In a study on the validity of appetite visual analogue scales, a Table was presented in which sample sizes were given in relation to detectable differences and power {Flint, 2000 #12}. Using this information, 18 subjects would be sufficient to detect a 10% difference in satiety, with 80% power at the 0.05 significance level for a paired design.”

 The results should be discussed with major argumentations.

Authors response: Thank you, we have expanded our argument that a smaller amount of potato could be used to generate equivalent feelings of satiety by adding the following: “This effect has been observed among an older group consuming an equivalent amount of carbohydrate either as a glucose beverage, instant potato or barley {Kaplan, 2002 #23}. All 10 participants were able to ingest the beverage but four and five of the participants were unable to eat all of the potato and barley, respectively {Kaplan, 2002 #23}. Despite the differences in carbohydrate intake of the subset who could not finish their food, as a group mean satiety was greater after potato than after the glucose beverage.”

 The authors should mark that this represents a pilot study.

Authors response: A pilot study is defined by the Collins English Dictionary as ‘a small-scale experiment or set of observations undertaken to decide how and whether to launch a full-scale project’. This study was adequately powered to detect the difference of interest (please see the additional information given in the Methods), so it does not fit the definition of a pilot study, it is a full-scale experiment in its own rite.

 The linguistic revision of whole manuscript is needed.

In details:

lines 29-30 should be rewritten; and lines 30-32 should be clarified:

Authors response: Thank you, we have clarified and added another reference as follows:  “Overweight and obesity are multifactorial in aetiology with such factors as the food environment, decreased physical activity, inadequate sleep and medication being involved {Wright, 2012 #2;Wilding, 2001 #3}. Highly palatable foods that are hard to resist eating are said to be ‘hyperpalatable’ and are characterised as being inexpensive, highly caloric, fat-laden, potentially addictive and a major contributor to chronic weight gain {Lerma-Cabrera, 2016 #4}{Gearhardt, 2011 #25}.”

 lines 34-37 should be rewritten;

Author response: We have expanded and clarified the sentence to the following: “However, the effectiveness of adding fruit and vegetables into diets as a means of reducing overall dietary energy density is questionable as a meta-analysis of the body of literature indicated no long-term effect on weight loss {Kaiser, 2014 #7}. This may be due to foods such as non-starchy vegetables having a low impact on feelings of satiety when eaten in typical amounts {Gustafsson, 1993 #26}. The authors of the meta-analysis concluded that increasing the fruit and vegetable intake of a diet is unlikely to be a successful weight control strategy because people tend not to decrease their overall dietary energy intake, suggesting that mechanisms underlying participant perceptions of hunger and satiety need to be better understood {Kaiser, 2014 #7}.”

 lines 30-44 these lines should be  better linked among them;

Author response: We have added the following as a better link: “The consumption of low energy dense foods must be accompanied with a concomitant reduction in the intake of high energy dense foods in order for an effect on body weight to be observed {Kaiser, 2014 #7}. Indeed, the energy density of diets, characterised by a combination of lower fat and higher fruit and vegetable intake, have been associated with long-term weight loss {Flood, 2009 #9}.”

 lines 45-48 should be enlarged.

Author response: We have added the following: “When foods were consumed in servings containing 1000kJ, satiety was inversely related to the energy density of the food, with a strong positive correlation found between the satiety index of food and the serving weight (r = 0.66, p < 0.001) (Holt). The least satiating foods were energy-dense snacks, confectionary and high-fat bakery products; among the starchy foods, white rice and pasta had satiety indices 1.19 and 1.38 times higher than white bread, with potato having a satiety index over three times (3.23) that of white bread {Holt, 1995 #11}.”

 The aim of work should be clarified.

Author response: Thank you, we have clarified this in the revised version of the latter part of the Introduction.

 Subparagraph 2.1 should be clarified and rewritten.

Author response: We have rewritten this as follows: “A randomized crossover design was used. On three different days, participants consumed a lunchtime meal comprising minced beef in a Bolognese sauce, mixed vegetables, and a serving of either whole boiled potato, white rice or penne pasta. The order in which participants received the meals was randomized and there was a minimum two-day washout between meals. Participants were asked to keep breakfast consistent on each of the three test days, to avoid consuming any food between breakfast and lunch, and to avoid any food intake for three hours after finishing the test meal, the period during which satiety was measured.”

 The authors should explain the choice of typology of test meal.

Author response: Our choice and rationale have been expanded on in the new sections of the Introduction, we hope this has been clarified.

 The data in table 3 should be better described

Author response: We have clarified Table 3 in the areas highlighted in yellow.

 The data in Figure 2 should be better described.

Author response: Satiety VAS data are commonly presented in this way. Additionally we have added to the footnote how to interpret the data as highlighted in yellow.

 In the discussion the data should better compared  with previous data in literature

Author response: We have added in a section expanding the comparison of our data with that of others as follows: “Our findings may be compared with a study in which investigators tested the effects on satiety of consuming meals containing various starchy staples {Geliebter, 2013 #24}. In that study, desire to eat was lower following baked potato compared with pasta but contrary to our findings, hunger and fullness AUC did not differ among meals containing instant potato, baked potato, brown rice, pasta and white bread {Geliebter, 2013 #24}. A difference between study designs was that Geleibter et al standardised the total water content of the meals (food plus water provided as a beverage) whereas we only standardised the volume of water given as an accompanying beverage. Thus, the meals and beverages that Geleibter et al provided presumably tended to be equi-volumetric {Geliebter, 2013 #24}. This may have obscured a difference in satiety among the meals given that food volume is a determinant of satiety {Rolls, 1998 #27}. This is suggestive that if starchy foods are to be exchanged, keeping accompanying drink volume constant may be important when assessing effects on satiety.”

 Round  2

Reviewer 2 Report

The authors have improved the manuscript.